# SAME MODEL, BETTER PERFORMANCE: THE IMPACT OF SHUFFLING ON DNA LANGUAGE MODELS BENCHMARKING

## ABSTRACT

Seemingly minor implementation details can significantly compromise benchmark validity. We demonstrate this through BEND (Benchmarking DNA Language Models), where hardware-dependent hyperparameters – number of data loading workers and buffer sizes – create spurious performance variations of up to 4% for identical models. The problem stems from inadequate data shuffling interacting with domain specific data characteristics. Experiments with three DNA language models (HyenaDNA, DNABERT-2, ResNet-LM) show these artifacts affect both absolute performance and relative model rankings. We propose a simple solution: pre-shuffling data before storage eliminates hardware dependencies while maintaining efficiency. This work highlights how standard ML practices can interact unexpectedly with domain-specific data characteristics, with broader implications for benchmark design in specialized domains.

## 1 INTRODUCTION

Standardized benchmarks serve as the foundation for scientific progress in machine learning, enabling researchers to compare methods, track improvements, and identify promising research directions. However, implementation of robust benchmarks that accurately reflect model capabilities without being affected by incidental parameters of the benchmarking framework, while remaining practical for widespread adoption, is challenging. Implementation details that appear benign can introduce subtle biases, create dependencies on computational resources, or favour certain approaches over others, ultimately compromising the benchmark's ability to provide fair and meaningful comparisons.

These challenges are particularly acute in emerging domains, like genomics, where domain-specific knowledge is scarce and benchmark design principles are still being established. The unique characteristics of biological data, such as spatial dependencies, sequence overlap, and domain-specific preprocessing requirements, can interact with standard machine learning practices in unexpected ways. As DNA language models (LMs) gain prominence for tasks ranging from regulatory element prediction to evolutionary analysis, the need for reliable evaluation frameworks becomes increasingly critical.

BEND (Benchmarking DNA Language Models) Marin et al. (2024) represents an important effort to standardize evaluation in this domain, providing a comprehensive suite of supervised genomic tasks, including CpG methylation prediction, histone modification annotation, chromatin accessibility, gene finding, and enhancer annotation. Like many modern benchmarks, BEND employs sophisticated data loading mechanisms to handle large-scale datasets efficiently, storing and streaming DNA sequence embeddings through a two-level shuffling strategy operating on dataset shards and sample buffers.

Here, we show that practical implementation choices made in BEND inadvertently influence benchmark results. The framework introduces dependencies on hardware-specific hyperparameters such as the number of data loading workers and buffer sizes. When combined with the inherent characteristics of genomic data, particularly the significant overlap between consecutive DNA sequence samples, these choices can lead to inadequate data shuffling and biased training dynamics. The choice of these parameters will likely correlate with the computational resources available to the

researchers and the dimension of embeddings of the model being evaluated. As a consequence, the results of BEND are biased to favour better resourced researchers, and introduce complex biases towards different model architectures.

We demonstrate that a simple pre-shuffling approach can eliminate these dependencies without changing BEND implementation details, while maintaining or improving performance across all tasks.

Our work contributes to the broader conversation about benchmark design best practices by providing a concrete example of how following a simple best practice avoids implementation artifacts that compromise evaluation validity. This discussion is particularly relevant as machine learning expands into specialized domains, where standard practices may interact with domain-specific characteristics in unforeseen ways.

## 2 BACKGROUND

### 2.1 BEND TASKS AND DATASETS

BEND evaluates the understanding of Language Models (LMs) of different DNA functional elements on a set of seven supervised and unsupervised tasks. Annotation data, used to extract DNA sequences from a reference genome, is provided for each task as *.bed* file, paired with the ground truth labels of the task. To evaluate a LM model on a specific task, the model embeds the task's DNA sequences, and the produced embeddings are evaluated. To allow for evaluation across LMs with different context window sizes, the length of the DNA sequences depends on the task configurations and the specific annotation, and DNA sequences that are longer than the LM's context window are split into chunks of a size supported by the model.

**Unsupervised tasks** are non-coding variant effect prediction for expression and disease in which single nucleotide mutations are classified as having an effect or not. Evaluation of the LM model is zero-shot and involves computing the cosine distance between the embeddings of the variant nucleotide and its reference nucleotide. The computed score is then compared to the ground truth labels.

**Supervised tasks** are CpG methylation, histone annotation, chromatin accessibility, gene finding and enhancer annotation, and involve finetuning a task specific LM prediction head with the remainder of the LM frozen. Each supervised task's data, except for enhancer annotation, is grouped by chromosome, sorted by the sequence start position, and split into train, validation, and test sets. The data of the enhancer annotation task is randomly shuffled and evaluated using cross-validation. The following steps summarise the supervised task pipeline:

1. **Embedding generation.** DNA sequences are extracted from a reference genome and embedded using the LM to evaluate.
2. **Training the downstream model.** The generated embeddings, paired with the relative labels, are used to train the downstream model in a supervised fashion.
3. **Evaluation of the downstream model.** The downstream model processes the embedded DNA sequences of the test split and the model predictions are compared to the ground truth.

Hence, benchmark results are determined by the downstream model performance, which is dependent on the given input data: the LM embeddings. The assumption is that the more the LM understands DNA, the more informative its embeddings are, and hence the better the downstream model performance is. The following section explains how, in the BEND implementation, this assumption does not hold.

### 2.2 WEBDATASET FOR STORING, LOADING AND SHUFFLING EMBEDDINGS

All supervised tasks annotations, except for the enhancer annotation task, are stored in genomic order, that is to say, they are grouped by chromosome and sorted by the position of the starting basepair of the DNA sequence. Epigenetic marks, such as CpG methylation, typically exhibit a distance-dependent correlation structure. Furthermore, any data samples closer than the task-specific sample DNA sequence length will contain overlapping DNA sequences. That is to say, genomic data

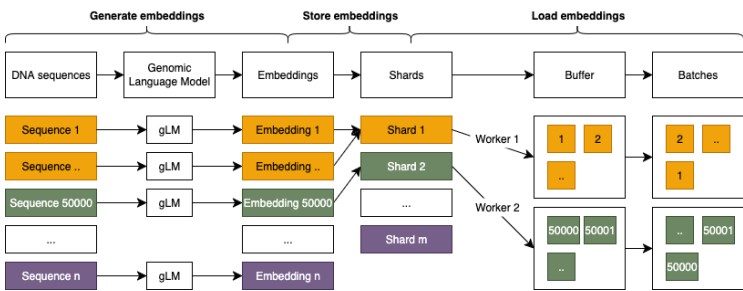

Figure 1: Pipeline for generating and storing and loading DNA sequences' embeddings using the WebDataset (Aizman et al., 2020) framework. First DNA sequences are sequentially embedded using the LM and stored into *shards*. Shards are assigned to workers and the number of workers is defined by hyperparameter. Workers load the embeddings from the shards into a buffer, which size is also a hyperparameter. Batches are created by randomly sampling embeddings from the buffer.

stored in genomic order is expected to exhibit large autocorrelation. Training on a datastream with high autocorrelation has a decremental effect on a stochastic optimisation algorithms, including AdamW (Loshchilov & Hutter, 2019), the optimiser used by BEND for finetuning the prediction heads. Shuffling breaks any correlations that would arise from data in genomic order and increases batch variety.

As the trunk of the evaluated LM is frozen, in order to avoid computing the same embeddings for each epoch, BEND computes the embeddings once and uses the WebDataset (Aizman et al., 2020) framework [1] to store, load and shuffle them from disk storage. WebDataset efficiently stores and iterates through a large dataset, without loading the entire dataset into RAM memory. This is achieved by storing the embeddings into *.tar* files called *shards*; when needed, it is possible to load batches of embeddings by reading shards and their content sequentially. Training data shuffling can be performed at three levels: shuffling sequences annotations before generating the embeddings, shuffling shards, and locally shuffling samples using a buffer.

Figure 1 shows the pipeline for storing, loading and shuffling the embeddings. In the BEND implementation, there is no shuffling step before storing data into shards. Shards themselves are implicitly shuffled when creating the dataloaders [2]. Hence, explicit shuffling is performed using a buffer, the size of which is a hyperparameter, where samples are shuffled before dividing them into batches. One shard is assigned to a single worker and each worker has its own buffer. Thus, the buffer shuffles only samples of the shards assigned to the worker. As the buffer loads samples into memory, the available memory is a bottleneck to increasing the buffer size. In BEND, the gene finding task uses only one worker and a buffer size of 1000 for 4783 training samples, leading to loading and shuffling one fifth of the training dataset before collecting it into batches of 64 samples. On the other hand, the CpG methylation task has a single worker and a buffer size of 200, which are not sufficient for shuffling a significant fraction of a training split of 959039 samples.

The order in which data are accessed is implicitly determined by the buffer size and another hyperparameter, the number of dataloaders. Figure 2 helps to understand sample access patterns across the entire dataset (first row) and the first batch (second row) for a number of representative scenarios. Specifically, No shuffle, BEND (1 worker), BEND (max workers), Pre-shuffle (1 worker), pre-shuffle (max workers) and Shuffle.

*No Shuffle* and *shuffle* are hypothetical cases in which data is accessed sequentially, or at random, as listed in the annotation files. In case of *no shuffle*, as shown in the first row, the first sample to be access has index 0, the second sample to be access has index 1, and so on. Thus, the first batch of size 256, will contain the first 256 samples. In *shuffle*, any sequence in the dataset could be accessed at any point, thus the first batch will contain a random distribution of sample indexes.

---

[1]https://github.com/webdataset/webdataset

[2]BEND implementation of WebDataset dataloaders:
https://github.com/frederikkemarin/BEND/blob/main/bend/utils/data_downstream.py

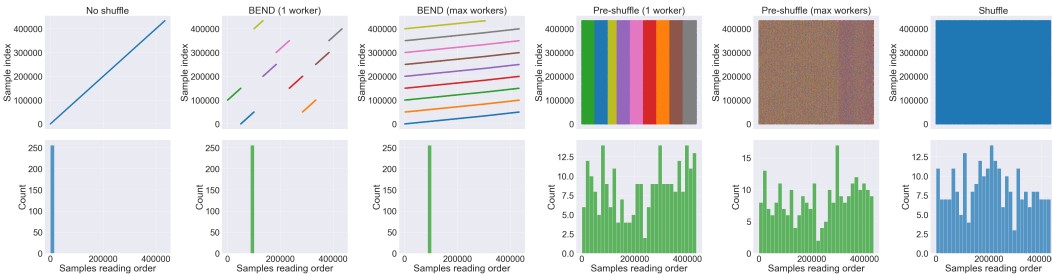

Figure 2: Impact of different approaches on the reading order of the histone modification task sequences: No shuffle, BEND (1 worker), BEND (max workers), Pre-shuffle (1 worker), pre-shuffle (max workers), Shuffle. The top row shows the impact of dataloaders in the sequences' access patterns across the entire dataset. The second row shows the impact of pre-shuffling on batch variety in terms of sample indexes. Different shards are depicted by different colours.

*BEND* and *pre-shuffle* illustrate the case in which WebDataset is used to load the histone modification embeddings using one or the maximum number of workers, which in this case is 9 as there are 9 shards. When using only 1 worker, shards are accessed one at a time, and read sequentially. Having multiple workers leads to composing multiple batches in parallel. Consequently, compared to *BEND (1 worker)*, *BEND (max workers)* improves sample variety between sequential batches, but it fails to affect sample variety within batches (see also A.1).

*Pre-shuffling* allows to store samples from any part of the dataset in any shard. In the *pre-shuffle* case, changes in the number of workers will only have an effect on performance. Indeed, as seen in the second row, *pre-shuffling* increases with-in batch variety and makes sample variety across consecutive batches independent of the number of workers.

In summary, in the BEND implementation shuffling is dictated solely by the buffer size and the number of dataloaders, failing to thoroughly shuffle the data. In addition, the buffer size and the number of dataloaders greatly vary between task configurations, and will be further dependent on the available computational resources.

### 2.3 OUR CONTRIBUTIONS

We add the missing step to the BEND pipeline: shuffling data annotations, to which we will refer as *pre-shuffle*.

We demonstrate the impacts on performance of the choice of number of workers and buffer size on the HyenaDNA-tiny-1k (Nguyen et al., 2023) model by:

- Comparison of results from the gene finding task with a buffer of size 0, instead of a buffer size of 1000 samples.
- Comparison of the results of the histone modification task using one worker, instead of 9 workers.
- Comparison of the results of the CpG methylation task using 15 workers, which is the number of training shards, instead of 1 worker.

Finally, we evaluate the impact of pre-shuffling the CpG methylation task data on two additional LM architectures, DNABERT-2 (Zhou et al., 2023) and ResNet-LM, a baseline model proposed by Marin et al. (2024).,

### 3 RESULTS AND DISCUSSION

#### 3.1 IMPACT OF HYPERPARMETERS

Table 1 demonstrates the impact of hyperparameters (number of workers and buffer size) on the benchmark results of the same model, HyenaDNA-tiny-1k (Nguyen et al., 2023).

Table 1: Test results of HyenaDNA-tiny-1k (Nguyen et al., 2023) on the CpG methylation and histone modification tasks using the minimum (1) and maximum amount of workers (with workers number equal to the shard numbers). Additionally, the HyenaDNA-tiny-1k was tested on the gene finding task using a buffer size of 1000 samples and without using the buffer.

| | CpG methylation (AUROC) | | Histone modification (AUROC) | | Gene finding (MCC) | |
|---|---|---|---|---|---|---|
| | Max workers | 1 worker | Max workers | 1 worker | 1000 size buffer | No buffer |
| BEND | 0.878 | 0.868 | 0.766 | 0.756 | 0.115 | 0.076 |
| Pre-shuffle | 0.901 | 0.900 | 0.772 | 0.771 | 0.120 | 0.116 |

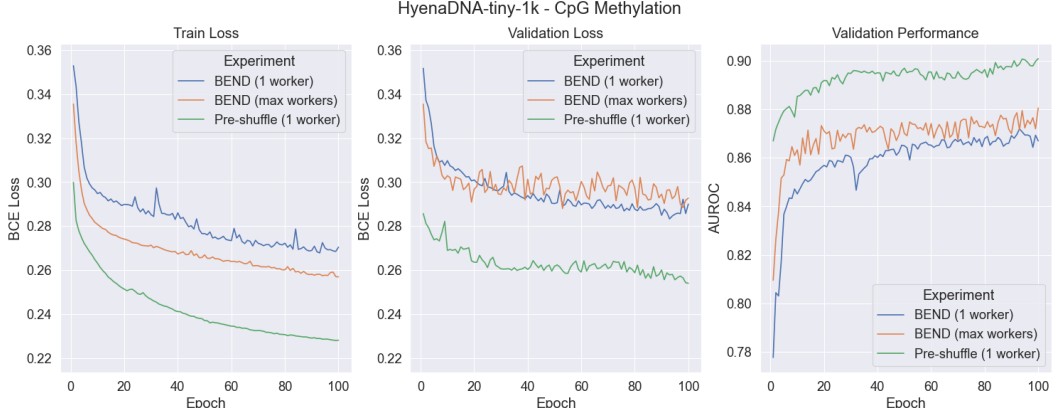

Figure 3: Training loss, validation loss and validation performance curves of the HyenaDNA-tiny-1k model on the CpG methylation task using three different approaches: pre-shuffle, BEND (1 worker) and BEND (max workers).

In both the CpG methylation and histone modification tasks, using a single worker decreases performance by 1% compared to using 15 and 9 workers, respectively.

Similarly, not using a buffer to shuffle samples before dividing them into batches leads to a loss of $\sim 4\%$ in performance on the gene finding task.

Pre-shuffling achieves comparable results independently of the hyperparameters (number of workers and buffer size) used.

## 3.2 CPG METHYLATION TASK GREATLY BENEFITS FROM SHUFFLING

The performance of BEND against pre-shuffling is comparable in the histone modification, as uses the max workers by default, and gene finding tasks, as the buffer size is set to 1000 by default. A comparison between BEND and pre-shuffle across tasks, using the default BEND hyperparameters is shown in the Appendix Figure 5.

However, using the default number of workers, which is 1, pre-shuffling increases the CpG methylation performance by 4% compared to BEND. Figure 3 shows the impact of pre-shuffling on the training loss, validation loss, and validation performance across epochs.

The reasons for this increase in performance could be the high autocorrelation in the DNA sequences of the CpG methylation task. For each task, we computed how many consecutive sequences overlap and the median percentage of overlapping length (see Appendix Table 3). In the CpG methylation task 51.9% of consecutive sequences overlap by at least one nucleotide. The median of the overlapping length is of 449 nucleotides, equal to 87.7% of the entire sequence length.

Table 2: Test results (AUROC) of the CpG methylation task using HyenaDNA-tiny-1k (Nguyen et al., 2023), DNABERT-2 Zhou et al. (2023), ResNet-LM (Marin et al., 2024) models.

|  | HyenaDNA-tiny-1k | DNABERT-2 | ResNet-LM |
|---|---|---|---|
| BEND | 0.868 | 0.893 | 0.890 |
| Pre-shuffled | 0.900 | 0.910 | 0.919 |

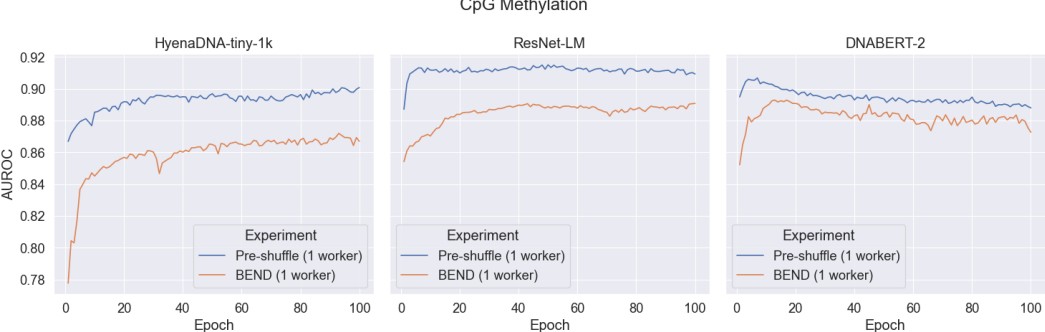

Figure 4: Validation performance of the HyenaDNA-tiny-1k (Nguyen et al., 2023), DNABERT-2 (Zhou et al., 2023) and the ResNet-LM (Marin et al., 2024) on the CpG methylation task, with and without pre-shuffling the annotation data.

To verify that the increase in performance on the CpG methylation is not unique to the HyenaDNA architecture, we replicated the experiment on DNABERT-2 Zhou et al. (2023) and ResNet-LM, a baseline model proposed with BEND (Marin et al., 2024). The experiment confirms that pre-shuffling annotation data of the CpG methylation task increases performance across different underlying model architectures (Table 2). Furthermore, when running the task without pre-shuffling, DNABERT-2 (Zhou et al., 2023) and ResNet-LM (Marin et al., 2024) models achieve comparable results, and both perform better than HyeanDNA-tiny-1k by about 2%. After pre-shuffling, ResNet-LM (Marin et al., 2024) becomes the best performing model, with an almost 1% increase in performance over DNABERT-2 and 2% increase over HyenaDNA-tiny-1k.

Shuffling in BEND is dependent on hyperparameters that are chosen based on the available resources, leading to complex biases. For example, buffer size is correlated to the available memory size and the LM embedding size. Hence, with fixed available memory, researcher would decrease the buffer size when benchmarking larger models which have embeddings occupying more memory individually, and with fewer samples loaded in the buffer results would be suboptimal. Conversely researchers could prioritise increasing the number of workers for larger models, as it is slower to load larger embeddings, leading to better performances.

## 4 CONCLUSION

We have demonstrated that hyperparameters that are often chosen based on computational resources, such as number of workers and buffer size, inadvertently affect benchmarking results. This is due to the BEND implementation for on-the-fly shuffling of the input data during prediction head finetuning being sensitive to the choice of these hyperparameters.

While we show that models independent of backbone architecture benefit from proper shuffling, with performance increases of up to 4%, these increases are not uniform. In fact, evaluating three dissimilar LM architectures (HyenaDNA-tiny-1k, DNABERT-2 and ResNet-LM) on the CpG methylation task leads to different conclusions when comparing proper and improper shuffling.

Finally, we have shown that pre-shuffling the data is a simple fix for disentangling benchmark performance from hardware-specific hyperparameters. We hope this paper provides a practical example highlighting the difficulty of properly implementing benchmarking frameworks, the need for

apprechiating domain specific knowledge and best practice, and more broadly the importance of following best practices and the unintended consequences when they are not respected.

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

## A  APPENDIX

| Task | Pct. of overlapping sequences | Median pct. of shared nucleotides | Weighted pct. of overlapping nucleotides |
|---|---|---|---|
| Histone modification | 17.03 | 19.92 | 3.39 |
| Gene finding | 7.09 | 12.39 | 0.88 |
| Enhancer annotation | 1.75 | 49.27 | 0.86 |
| CpG methylation | 51.88 | 87.70 | 45.50 |
| Chromatin accessibility | 28.29 | 20.31 | 5.75 |

Table 3: For each task, pct. of consecutive overlapping DNA sequences, median pct. of shared nucleotide across overlapping sequences and a weighted pct. computed by multiplying the previous percentage types.

### A.1  IMPACT OF DATALOADERS AND BUFFER ON DATA ACCESS ORDER

Figure 6 shows histone modification task data access order using different approaches: *No shuffle*, *BEND (1 worker, no buffer)*, *BEND (max workers, max buffer)*, *pre-shuffle (1 worker, no buffer)* and *shuffle*. All approaches, except for *BEND (max workers, max buffer)* are explained in Section 2.2. The *BEND (max workers, max buffer)* includes the use of the WebDataset (Aizman et al., 2020) buffer of size equal to the number of samples in a shard, which in the case of BEND is 50,000. As

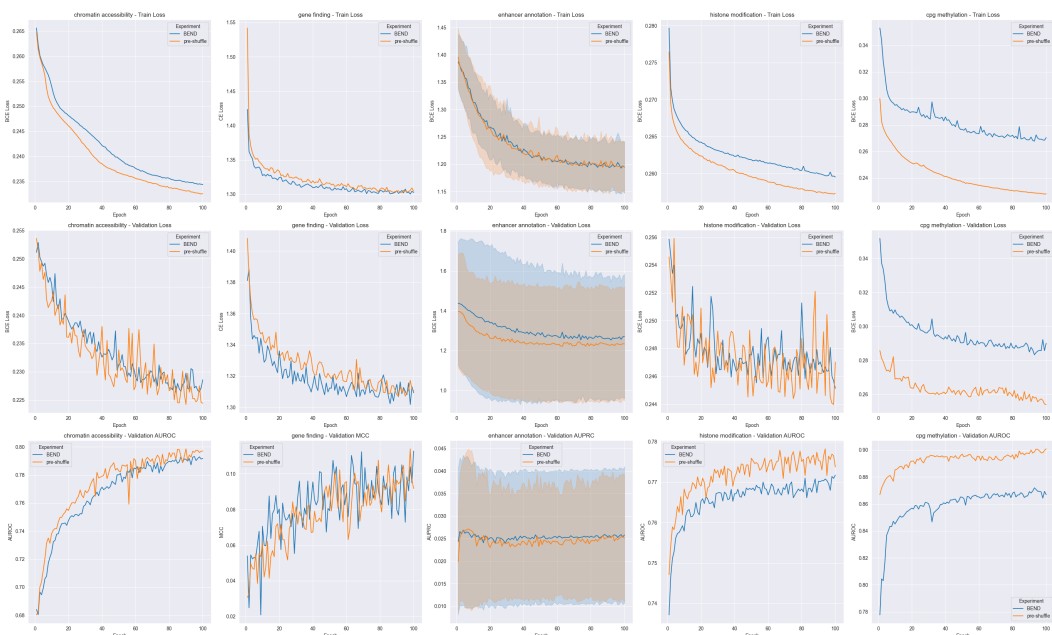

Figure 5: Comparison on all tasks of the BEND (Marin et al., 2024) pipeline with and without pre-shuffling using the HyenaDNA-tiny-1k (Nguyen et al., 2023) model. For the enhancer annotation task, it is displayed the mean performance across folds and the standard deviation.

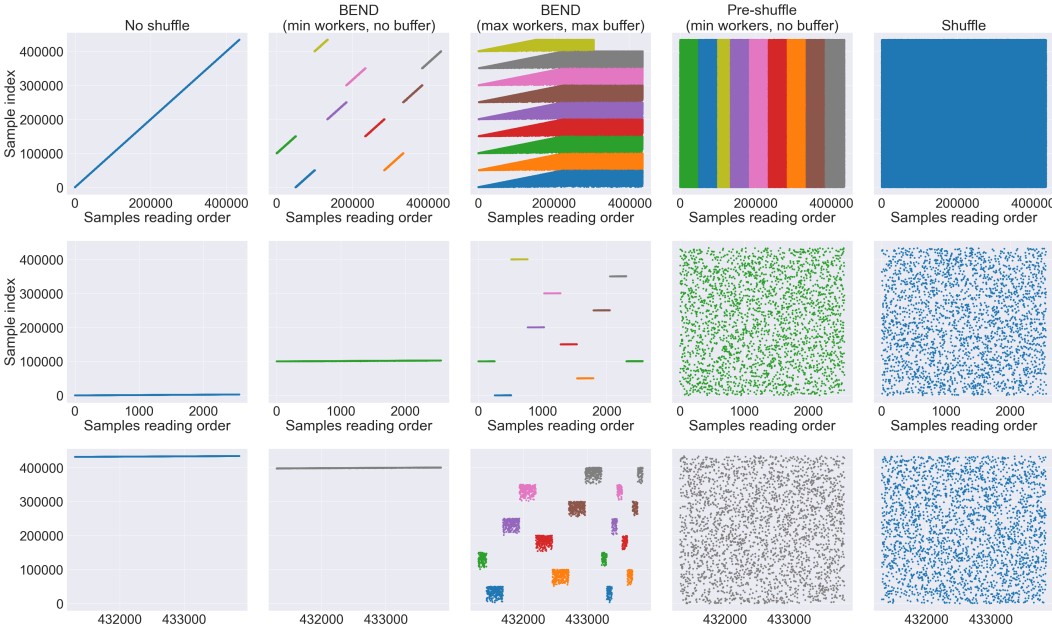

Figure 6: Impact of different approaches on the reading order of the histone modification task sequences: No shuffle, BEND (1 worker, no buffer), BEND (max workers, max buffer), Pre-shuffle (1 worker, no buffer), Shuffle. The first row shows the impact of dataloaders and buffer in the sequences' access patterns across the entire dataset. The second row shows the batch variety, in terms of sample indexes, of the first ten batches. Finally, the third row shows the batch variety of the last ten batches. Different shards are depicted by different colours.

seen in the second row of the *BEND (max workers, max buffer)* approach, initially, only samples at the beginning of the shard are accessed, as batches are composed while samples are loaded into the buffer. However, over time, more samples are loaded than accessed, filling the buffer. This leads to batches having samples from any part of the shard, as seen in the plot in the last row.

