# OpenReview forum: "Same model, better performance: the impact of shuffling on DNA Language Models benchmarking"
_ICLR.cc/2026/Conference — Submitted to ICLR 2026_

### Official Review · Reviewer_6TxJ · 2025-10-16

**Soundness:** 2
**Presentation:** 2
**Contribution:** 2
**Rating:** 2
**Confidence:** 3

**Summary:**

The paper audits BEND, a benchmark for DNA language models, and argues that seemingly innocuous dataloader choices (number of workers, buffer size) leak into evaluation by weakening shuffling and amplifying genomic autocorrelation. Empirically, the authors show that:

(i) fewer workers and small buffers reduce AUROC/MCC in several BEND tasks for the same model;

(ii) a simple “pre‑shuffling” of training annotations (before writing WebDataset shards) removes the hardware–hyperparameter dependence and usually improves scores;

and (iii) the model ranking on CpG methylation can flip after proper shuffling.The remedy is to shuffle annotations prior to storage so that within‑batch variety no longer depends on workers/buffer.

**Strengths:**

1. The author demonstrate the phenomenon across different backbones, proving that the problem exists widely and is caused by the dataloader instead of model difference.

2. The paper quantifies sequence overlap per task, explaining why CpG methylation is most sensitive. This explaination further explain the phenonemon.

3. The pre‑shuffling restores within‑batch diversity irrespective of worker count and buffer size is simple enough to demonstrate a raise in the BEND benchmark.

**Weaknesses:**

1. Insufficient statistical rigor. No seeds, variances, or statistical tests. Even training curves lack shading for variability. Given small absolute gains, this could be the possible reason for the nuiance difference of the models.

2. Narrow novelty. The fix is a well‑known practice in general domain in learning thoery. Without a more systematic characterization (e.g., formal mixing analysis or a generic recipe for streaming datasets with spatial autocorrelation), the contribution feels more like a benchmark note than a full conference paper.

**Questions:**

1. Variance and replicability. Report mean with std over 3 seeds for all entries in Tables 1–2 and the Fig. 5 task panel. How stable are the gains and the ranking changes?

2. Broader applicability. Evaluate at least one non‑BEND genomics dataset (or a synthetic autocorrelated dataset) to support the claim that this is a general issue in specialized domains, not a BEND‑specific implementation artifact.

---

### Official Review · Reviewer_TSpt · 2025-10-31

**Soundness:** 3
**Presentation:** 2
**Contribution:** 1
**Rating:** 2
**Confidence:** 5

**Summary:**

This paper identifies a critical implementation flaw in the BEND (Benchmarking DNA Language Models) framework, where hardware-dependent hyperparameters (number of data loading workers and buffer sizes) create spurious performance variations of up to 4% for identical models. The authors demonstrate that inadequate data shuffling, when combined with genomic data characteristics (high sequence overlap and spatial autocorrelation), leads to biased training dynamics. They propose pre-shuffling data before storage as a solution and validate this across three DNA language models (HyenaDNA, DNABERT-2, ResNet-LM).

**Strengths:**

1. The paper identifies a real implementation problem in an existing benchmark that could affect reproducibility and fair model comparison, which is valuable for the genomics ML community.
2. The authors provide concrete evidence showing how hyperparameters affect performance across multiple models and tasks, with particularly compelling results on the CpG methylation task (4% improvement)

**Weaknesses:**

1. This work is essentially a bug report and fix for a specific benchmark implementation rather than a novel research contribution. While valuable for practitioners, it lacks the theoretical depth, methodological innovation, or generalizable insights expected of a venue like ICLR. The core finding that data should be properly shuffled is a well-established ML best practice, not a research finding.
2. Only evaluates on BEND tasks; no exploration of whether similar issues exist in other genomic or biological sequence benchmarks
3. The main body of the paper contains only 6 pages of actual content, which clearly falls short of the research depth and comprehensiveness expected for an ICLR proceedings paper.

**Questions:**

1. Why not compare against simply using standard PyTorch DataLoaders with proper shuffling instead of WebDataset

---

### Official Review · Reviewer_1AvC · 2025-10-31

**Soundness:** 3
**Presentation:** 1
**Contribution:** 2
**Rating:** 2
**Confidence:** 4

**Summary:**

The manuscript addresses the issue of high batch homogeneity that arises when running tasks from the BEND benchmark, stemming from the collision of specific technical choices in the Webdataset dataloader provided by the original authors and autocorrelation in DNA sequences stored in genomic order. This issue adversely affects model performance, and the authors convincingly demonstrate that introducing a simple shuffling step substantially improves results across multiple architectures. This finding is particularly noteworthy, specifically for the DNA modelling space, as such effects may not be immediately evident when working with DNA data, given its inherent complexity and high dimensionality.

**Strengths:**

1.	The issue identified with the WebDataset dataloader is described in thorough technical detail and effectively connected to theory
2.	The authors present a well-designed experiment with clear, reproducible results demonstrated across multiple models.
3.	The observed performance gains are substantial enough to potentially affect existing benchmarks and the relative ranking of models, with one example provided by the authors.

**Weaknesses:**

I do not want the authors to be discouraged, this is a clever piece of work that highlights an important issue, one that even prompted me to revisit aspects of my own work. Its relevance is clear. However, I do not believe the paper has yet reached the threshold for a main track conference publication although I do think they should contact the authors of the original BEND benchmark. As it stands, the manuscript focuses on a subset of one benchmark, which limits its scope. If extended to other benchmarks that do not employ WebDataset, I would expect a reasonably skilled ML practitioner to shuffle their data by default. In the case of the Webdataset I suspect everybody is using it in the same way If the authors believe this assumption does not hold, I would have appreciated further commentary or evidence addressing whether this is likely to be wider spread. I would also suggest to the authors to focus on enhancing clarity in their plots and captions (e.g. Figure 2 is hard to read, and I believe the caption talking about the top and bottom row is misleading), and also revisit some sections from their manuscript (for example non-coding variants are mentioned in 2.1 but I could not find results for it)

**Questions:**

1. Can the authors highlight other benchmarks or tasks where this might be an issue? I do suspect there will be other DNA modelling benchmarks where these issues may arise
2. DNABERT-2, HyenaDNA and ResNet-LM involve substantially different modelling choices (convolutions vs attention, BPE tokenization vs single nucleotide). While the autocorrelation is probably affecting all types of models I suspect they are not all affected equally as the results suggest. An in-depth treatment of these issue can substantially increase the scope of the manuscript.
3. Would be expect these issues to arise in a self-supervised setting/foundation model training?

---

### Official Review · Reviewer_c9hF · 2025-11-01

**Soundness:** 2
**Presentation:** 2
**Contribution:** 1
**Rating:** 0
**Confidence:** 4

**Summary:**

This paper highlights some issues with the data shuffling of the BEND benchmark due to the implementation of it. Briefly, the BEND dataset on WebDataset is stored in genomic order (per shard), and then loading and shuffling the data the dataset is not shuffled globally, leading to issues downstream due to autocorrelation between samples within shards (due to being from similar genomic regions). This paper highlights these issues, and presents a simple fix (shuffle the dataset before you store it).

**Strengths:**

This paper identifies a real tangible problem and poses a very simple fix for it. I'd sincerely like to thank the authors for identifying and correcting this. Issues with datasets exactly like this are known to cause problems, and even though the fix is quite simple, actually fixing issues like this are of great interest to our field.

**Weaknesses:**

The impact of shuffling (or lack thereof) is a known problem in ML so the results are unsurprising. While identifying and fixing issues such like is of paramount importance, there is no real academic novelty and further I don't believe this is in scope for publication in ICLR.

**Questions:**

N/A, I believe most of the salient points were presented clearly. If the authors can provide some compelling evidence that their work is actually in scope, I would be willing to revise this review.

---

### Meta-Review · Area_Chair_R7Lf · 2026-01-05

**Summary:**

This paper analyzes an implementation issue in the BEND benchmark, showing that insufficient data shuffling in WebDataset can lead to hardware-dependent performance differences due to genomic autocorrelation. The authors demonstrate that a simple pre-shuffling step largely resolves this issue and improves reported results across several models.

All reviewers agree that the paper identifies a real and practically relevant problem, and the empirical analysis is clear. However, there is also strong consensus that the contribution is limited in scope and primarily amounts to a benchmark audit or bug fix. Proper data shuffling is a well-known best practice in machine learning, and the paper does not introduce new methodology, theory, or generalizable insights at the level expected for ICLR. The evaluation is also restricted to a subset of a single benchmark, further limiting its impact.

Given the lack of rebuttal and the consistent reviewer feedback regarding scope and novelty, I recommend rejection. The work is nonetheless valuable for the community and would be better suited for a benchmark, reproducibility, or dataset-focused venue.

**Reviewer Concerns:**

Since there was no rebuttal, the reviewers concerns remain.

**Reviewer Scores:**

Since there was no rebuttal, the reviewers' scores would remain.

---

### Decision · Program_Chairs · 2026-01-26

Reject